# *FLTX2*: A Novel Tamoxifen Derivative Endowed with Antiestrogenic, Fluorescent, and Photosensitizer Properties

**DOI:** 10.3390/ijms22105339

**Published:** 2021-05-19

**Authors:** Mario Díaz, Fernando Lobo, Dácil Hernández, Ángel Amesty, Catalina Valdés-Baizabal, Ana Canerina-Amaro, Fátima Mesa-Herrera, Kevin Soler, Alicia Boto, Raquel Marín, Ana Estévez-Braun, Fernando Lahoz

**Affiliations:** 1Departamento Biología Animal, Edafología y Geología, Universidad de La Laguna, 38200 Tenerife, Spain; fatimamesaherrera@gmail.com; 2Unidad Asociada ULL-CSIC “Fisiología y Biofísica de la Membrana Celular en Enfermedades Neurodegenerativas y Tumorales”, 38200 Tenerife, Spain; alicia@ipna.csic.es (A.B.); rmarin@ull.edu.es (R.M.); flahoz@ull.es (F.L.); 3Programa Agustín de Betancourt, Universidad de la Laguna, 38200 Tenerife, Spain; flobopal@ull.edu.es (F.L.); angelamesty@yahoo.es (Á.A.); cvaldesb@ull.edu.es (C.V.-B.); 4Instituto de Productos Naturales y Agrobiología del CSIC, Avda. Astrofísico F. Sánchez, 38206 Tenerife, Spain; dacil@ipna.csic.es; 5Instituto Universitario de Bioorgánica “Antonio González”, Universidad de La Laguna, 38200 Tenerife, Spain; aestebra@ull.edu.es; 6Departamento Ciencias Médicas Básicas, Universidad de La Laguna, 38200 Tenerife, Spain; anacanerinaamaro@gmail.com; 7Departamento Física, IUdEA, Universidad de La Laguna, 38200 Tenerife, Spain; kscarracedo@gmail.com; 8Departamento Química Orgánica, Universidad de La Laguna, 38200 Tenerife, Spain

**Keywords:** tamoxifen, estrogen receptors, SERM, fluorescence, FRET, reactive oxygen species, superoxide anions, photosensitization, FLTX1, breast cancer, laser dye, molecular dynamics

## Abstract

Tamoxifen is the most widely used selective modulator of estrogen receptors (SERM) and the first strategy as coadjuvant therapy for the treatment of estrogen-receptor (ER) positive breast cancer worldwide. In spite of such success, tamoxifen is not devoid of undesirable effects, the most life-threatening reported so far affecting uterine tissues. Indeed, tamoxifen treatment is discouraged in women under risk of uterine cancers. Recent molecular design efforts have endeavoured the development of tamoxifen derivatives with antiestrogen properties but lacking agonistic uterine tropism. One of this is FLTX2, formed by the covalent binding of tamoxifen as ER binding core, 7-nitrobenzofurazan (NBD) as the florescent dye, and Rose Bengal (RB) as source for reactive oxygen species. Our analyses demonstrate (1) FLTX2 is endowed with similar antiestrogen potency as tamoxifen and its predecessor FLTX1, (2) shows a strong absorption in the blue spectral range, associated to the NBD moiety, which efficiently transfers the excitation energy to RB through intramolecular FRET mechanism, (3) generates superoxide anions in a concentration- and irradiation time-dependent process, and (4) Induces concentration- and time-dependent MCF7 apoptotic cell death. These properties make FLTX2 a very promising candidate to lead a novel generation of SERMs with the endogenous capacity to promote breast tumour cell death in situ by photosensitization.

## 1. Introduction

Tamoxifen (TX) is a selective modulator of estrogen receptors (SERMs) widely used as coadjuvant therapy in the treatment of breast cancer. Tamoxifen has been prescribed for decades to millions of women as the first therapeutic strategy for prevention or treatment hormone-dependent breast cancer [1,2,3,4]. TX is a triphenylethylene molecule that competes the binding of 17β-estradiol to estrogen receptors (ER) and prevents the transcriptional activation of estrogen-dependent genes, which control the proliferation of mammary gland cells. However, tamoxifen is known to significantly enhance the risk of developing endometrial lesions, including hyperplasia, polyps, carcinomas, sarcoma, as well as womb cancer and thromboembolism [5,6,7,8,9,10]. Notably, tamoxifen-associated endometrial cancer often has a poor clinical outcome, and serious concerns exist on its use for cancer prevention or long-term palliative treatment [9,11]. The current view of these undesired effects is that in most cases they are due to the interaction of TX with either canonical and/or non-canonical molecular targets [11], whose proper identification has been hampered because the lack of appropriate pharmacological and experimental tools. The existence of these adverse secondary effects has boosted many laboratories to search for either modifications on the tamoxifen molecule or novel SERMs structurally unrelated to tamoxifen, which could circumvent the undesirable effects during long-term therapies [1,12,13,14,15]. In recent years, we have developed a novel fluorescent tamoxifen derivative, FLTX1, which not only allows tracking canonical (i.e., ER) and non-canonical intracellular targets, but also retains most pharmacological properties of tamoxifen (specific ER binding, antagonism of ER-mediated transcriptional activation and cell proliferation), while lacks the trophic worrisome effects of tamoxifen in the uterus [16]. Of note, beyond its pharmacological properties, FLTX1 is endowed with emergent opto-chemical properties of which tamoxifen lacks [17,18,19,20]. In the present study we have designed and developed a new tamoxifen derivative, named FLTX2, formed by the covalent linkage of FLTX1 to a well-known photosentizer, Rose Bengal (RB), which takes advantages of both pharmacological and optochemical properties of FLTX1, and the incorporation of a novel functionality to generate reactive oxygen species (ROS) with high efficiency. Thus, positive gain media based on FLTX1 optical properties were prepared and laser emission was demonstrated under optical pulsed excitation, which is transferred to RB through Förster resonance energy transfer (FRET). In this article we have used a multidisciplinary approach to assess the main properties of FLTX2 with special focus on: (1) ER antagonistic properties, (2) Docking and Molecular Dynamics of the FLTX2-ER interaction (3) Optochemical properties, (4) Irradiation-induced ROS generation, (5) Intracellular binding, and (6) Ability to photosensitize and to induce irradiation-mediated cell death. The outcomes pinpoint to a highly promising pharmacophore with potential applications as specific site-directed photosensitizer in ER-dependent breast cancer.

## 2. Results and Discussion

### 2.1. Synthesis of FLTX2 (FLTX1-RB)

In the present study we have designed and synthetized a novel fluorescent tamoxifen derivative, FLTX2, following a synthetic strategy (Figure 1A) which avoided the need of high-cost commercial tamoxifen. The tamoxifen core of dimethyl acetal precursor **1** was synthesized in a few steps from inexpensive materials, according to a reported procedure [21]. As it happens with commercial tamoxifen, a mixture of the E and Z isomers was obtained. Once compound **1** was available, the aldehyde was deprotected in acid media and treated with ethanolamine under reductive conditions to introduce the FLTX1-Rose Bengal (FLTX1-RB) linker. This linker was attached first to 7-nitrobenzofurazan (NBD) and then to RB. Thus, the FLTX1-linker intermediate was treated with 4-chloro-7-nitrobenzofurazan in the presence of a base, providing the desired compound **2**. The **1→2** conversion took place in a satisfactory overall yield (29% for the four steps), since the procedure avoids lengthy protection-deprotection steps and is operationally simple. However, we expect to further optimize the yields in a future. In the final synthetic step (conversion **2→3**), substrate **2** underwent an esterification reaction with Rose Bengal under mild conditions, affording FLTX2 (**3**). A 3D representation of FLTX2 in a minimum-energy conformation is shown in Figure 1B, where it can be seen that the NBD and Rose Bengal moieties align closely allowing FRET exchange. With respect to the NBD-BR approach, it should be pointed out that the linker length (a two-carbon chain) was chosen as optimal according to our former study [22] aimed at assessing FRET efficiencies between the NBD-moiety of FLTX1 (donor) and RB (acceptor) dipole pairs (see below).

### 2.2. Affinity Studies of FLTX2 on ER. Comparison with FLTX1

Competition experiments were performed using rat uterus cytosol. This extract rich in ERα was saturated with 5 nM of labelled E2 in the presence of increasing concentrations of unlabelled tamoxifen or FLTX2 (Figure 2).

As expected, TX was able to competitively displace the [^3^H]E2 from rat uterine ER in a dose-dependent manner. Similarly, FLTX2 competed off the radiolabelled estradiol binding to ER. The estimated EC_50_ values for TX and FLTX2 were 173.3 ± 6.14 nM (*p* < 0.005) and 114.4 ± 3.07 nM (*p* < 0.05), respectively. Hill coefficients (n_H_) were close to 1 in both cases. Thus, assuming an RBA (Relative Binding Affinity) value of 100 for TX, the RBA value for FLTX2 was 151.5% indicating a slightly increased affinity of FLTX2 for ER than TX. Thus, it may be concluded that despite significant changes in the molecular complexity and size of FLTX2 compared to TX, the novel derivative preserves the ability to compete E2 for ERα binding, behaving as antiestrogen and likely with a slightly higher affinity than TX and FLTX1 [16]. These properties of fluorescent derivatives FLTX1, and especially FLTX2, were remarkable since other studies have shown that, in general, changes in the side chain of TX lead to compounds with decreased affinity for ER [23,24,25]. Nonetheless, from present results we cannot ascertain if FLTX2 behaves as a pure antiestrogen and whether it lacks the agonist functionality demonstrated for TX, in particular, in uterine tissues. However, in our previous study on FLTX1 pharmacological properties, we demonstrated that this pharmacophore was completely devoid of uterotrophic and proliferative effects [16]. In addition, FLTX1 was unable to induce ER-mediated transcriptional activity in transfected MCF7 and T47D-KBluc cells expressing a genetic construct containing a luciferase reporter gene under the control of canonical estrogen response elements [16]. These findings were interpreted in terms of a physical hindrance of FLTX1-bound ER to recruit coactivators required for transcriptional activation. Based on the chemical similarities between bulky side chains of FLTX1 and FLTX2, and especially from the results of docking and MD detailed below, it is likely that FLTX2 might also lack agonistic effects exhibited by TX.

### 2.3. Docking of FLTX2 on ERα. Comparison with FLTX1

In order to explain the increased affinity of FLTX2 for ERα compared to FLTX1 and TX, we carried out a molecular docking study on the reported crystal structure of human ERα ligand binding domain (LBD) in complex with 4-hydroxytamoxifen (PDB code 3ERT), which is an active metabolite of TX and a well-known as selective estrogen receptor modulator (SERM). The ligand-binding domain (LBD) is located in the middle of the carboxy-terminal region of the receptor, the final portion of which is critical and responsible for ligand binding, receptor dimerization, and nuclear translocation among other roles [26,27,28].

The LBD consists of twelve α-helices (H1 to H12) and a beta sheet/hairpin. The amino acid residues that line the ligand-binding cavity or interact with bound ligand span from helix 3 (H3) to helix 12 (H12). When the receptor binds to the ligand, a change in its three-dimensional structure is produced, and the LBD forms a bag-shaped structure, hydrophobic in nature, that lodges the ligand [29].

The H12 plays an important role as a molecular switch by adopting different conformations that allow ligand-dependent receptor activation [30]. When bound to an agonist, the LBD adopts an active conformation where H12 rests across H3 and H11, forming a groove to accommodate co-regulator binding and facilitate downstream activation process [31]. When bound to an antagonist, H12 is displaced from this position resulting in the distortion of the co-regulator binding groove and the inhibition of receptor activation [26,28,32,33].

Hence, in order to explore the binding mode of FLTX2 derivative, we have docked FLTX2, FLTX1, and TX into the hydrophobic binding pocket of LBD to understand their possible binding modes and key active site interactions (Figure 3).

The analyses of the docking results showed that the moieties of both derivatives (FLTX1 and FLTX2) share a similar pose at the ligand binding site and that it is also located towards the end of the pocket where the ligand is usually found in the crystal structure (Figure 3B,C). These results strongly suggested that these compounds share a common binding mode into the hydrophobic binding pocket of ERα (Figure 3D).

According to the predicted binding modes, FLTX1 as well as FLTX2 display the same type of hydrophobic interactions as tamoxifen does, therefore they probably play a dominant role in protein–ligand interaction. In the favored docking conformations, there are multiple potential hydrophobic interactions involving residues Leu391, Leu349, Leu346, Leu428, Leu387, Leu384, Leu525, Ile424, Glu353, Arg394, Met421, Met388, Met343, Gly420, Ala350, Thr347, and His524, whose side chains are in close proximity to the tamoxifen moiety (see Figure 3A–C).

Additionally, our results also show that the introduction of the NBD moiety to tamoxifen to form the derivative FLTX1, as well as the preparation of the derivative FLTX2 by modifying the derivative FLTX1 to form the derivative Rose Bengal-conjugated FLTX1, gives rise to a new hydrogen bond interaction with the residue Cys530 in H12. However, in the case of FLTX1 this interaction is produced by the nitro group (-NO_2_) of NBD while in the case of the derivative FLTX2 the interaction is produced by the hydroxyl group (-OH) present in Rose Bengal (Figure 3D).

The calculations revealed that the affinity for the receptor binding site. It is as high for both derivatives (XPGlide score of ×11.55 kcal mol^−1^ for FLTX1 and −11.34 kcal mol^−1^ for FLTX2) as for tamoxifen (XPGlide score −11.72 kcal mol^−1^), which was used as internal reference. These findings suggest that novel interactions observed for FLTX1 and FLTX2 with the Cys530 residue in H12 are as effective as that established by tamoxifen between the protonated nitrogen and Asp351 which provides a stabilizing effect on the active antiestrogenic conformation of ER-LBD [34]. 

In addition, due to the presence of both NBD and RB moieties in FLTX2, additional hydrophobic interactions arise with residues Leu539, Leu536, Val534, Val533, Asp351, Trp383, and Tyr526, which likely stabilize H12 in its typical antagonist conformation, thereby hampering coactivators to interact with the receptor, and more efficiently binding of FLTX2 to the RBS [28]. This issue is further explored in the next section.

### 2.4. Molecular Dynamic (MD) Simulation

To further evaluate the reliability of the docking study result, we combined it with more accurate molecular dynamics (MD) simulation techniques in order to predict more reliable structures as well as to confirm the stability of the system and study the conformational change in the ligand-receptor complex throughout the course of the MD simulation. The simulations provide every minute of detail on every atom movement with respect to time. This will help in answering the questions arising about the binding mode, stability, deviation, and fluctuation pattern of the protein.

The MD simulation was performed on the best Docking pose of the FLTX2 derivative for 50 ns in an explicit aqueous solution environment with periodic boundary conditions, the OPL-2005 force field and the TIP3P solvent model employing the Desmond simulation package seamlessly integrated into Maestro software. The simulation was also carried out to confirm the orientation of the FLTX2 derivative into the binding site.

The analysis of the trajectory obtained during the simulation as well as the protein–ligand contacts analysis revealed that the compound does not leave the binding site of the protein and remained in the similar orientation during the entire simulation. However, in the graphical snapshot of the production phase the disappearance of the hydrogen bond interaction formed between the hydroxyl group present in Rose Bengal with the residue Cys530 could be observed due to the flexibility of movement of H12, and it could also be detected by the appearance of two new hydrogen bonds mediated by two water molecules formed between the carbonyl group of RB and the residues Gly344 and Thr347. In addition, a previously undetected π–π stacking interaction between TX with the residue Phe404 could also be observed. This is indicative that FLTX2 is further stabilized by other hydrogen bonds, which could explain a significant increase in the FLTX2 affinity of this derivative (Figure 4A).

Additionally, the interaction observed during the simulation between the TX core of FLTX2 with Phe404 (Figure 4B) is quite significant since it is observed around more than 50% of the simulation time in the selected trajectory (Appendix A).

In addition, it could also be observed that the FLTX2 interacted with other essential amino acids such as Ala350, Leu525, and Ser527, which are part of the hydrophobic pocket by means of hydrophobic interactions (Figure 4B). These interactions of the protein with the ligand may be quantified as the ‘interaction fraction’ throughout the whole molecular dynamic simulation (Figure 4C and Appendix A). The stacked bar charts are normalized over the course of the trajectory (i.e., a value of 0.7 indicates that along 70% of the simulation time the specific interaction is maintained). This allowed us to know the role of each of the particular bonds, with the amino acid residues responsible for the stabilization of the complex.

This model and subsequent dynamical observation has notably expanded our knowledge on the potential antiestrogenic activity of FLTX2 on the ERα. Thus, FLTX2 retains the ability of TX to get into the ERα ligand binding groove and to establish similar interactions with the LBD though and with similar affinity likely through π–π stacking interaction with Phe404. However, the large moiety formed by NBD and RB expand outside the ligand groove and stabilizes H12 in the antiestrogen conformation, by the hydroxyl group present in Rose Bengal with the residue Cys530, and two hydrogen bonds with the carbonyl group of RB and the residues Gly344 and Thr347. In this sense, FLTX2 is expected to exhibit a ligand kinetics that tends to the formation of the ERα-FLTX2 complex, thereby behaving as an antagonist as effective as TX and FLTX1.

In our previous study on the pharmacological properties of FLTX1, we demonstrated the total absence of estrogen-like effects both in vitro and in vivo as one of the most outstanding properties of this pharmacophore [16]. The antiestrogenic ability of FLTX1 and FLTX2 is mechanistically similar to that of TX, i.e., by binding the LBD domain through their triphenylethylene cores and the lateral side moieties protruding out the LDB pocket and displacing helix 12 from adopting its agonist conformation [12,35]. Further, the compact volume of NDB-RB moiety and its spatial conformation respect to the triphenyethylene core as indicated by the minimal-energy conformations and molecular dynamics simulation (Figure 3 and Figure 4) strongly suggest a conformational hindrance for helixes 3 and 11 to reallocate helix 12 [12,35,36]. It is known that for agonist ER ligands, helix 12 is stabilized in a conformation that allows it to form one side of the coactivator-binding site [12,37,38,39]. The geometry of FLTX2-bound ER structure would hamper the sequential recruitment of tissue-specific coregulators, including steroid receptor co-activator-1 (SRC-1), amplified in breast cancer-1 (AIB1) and CREB-binding protein (CBP) which are required in endometrial cells for estrogenic agonism and carcinogenesis [11,28,37,40,41]. Future studies will be needed to confirm the pure antiestrogenic nature of FLTX2. 

### 2.5. Fluorescence Properties and FRET Mechanism

The visible optical absorption of FLTX1, RB, and FLTX2 solutions at a concentration of 10 µM in DMSO are given in Figure 5A.

The characteristic broad band centered at about 488 nm is observed for FLTX1 [18]. The absorption spectrum of RB shows an intense band at about 565 nm and a shoulder at 525 nm. The FLTX2 complex shows the absorption features of their constituent moieties. The absorption spectrum has a broad band at 488 nm, associated with the FLTX1 moiety, and also the absorption bands of RB but slightly red shifted (main band at 574 nm and shoulder at 532 nm). The red shift observed in the bands related to the RB moiety indicates that the covalent bonding between FLTX1 and RB changes the environment of the RB electronic density and has an influence on its energy levels. Similar effects have been reported for variations of the RB molecular environment [42].

The normalized fluorescence emission spectra of the FLTX1, RB, and FLTX2 solutions under excitation at 475 nm are given in Figure 5B. An intense and green emission band with a maximum about 550 nm is observed for the FLTX1. The RB emission is characterized by a broad red band with its maximum at about 608 nm. When FLTX2 is excited at 475 nm, the emission of the FLTX1 moiety is almost completely quenched and essentially the emission is only due to the RB moiety. However, upon 475 nm excitation the FLTX1 moiety of the FLTX2 complex is excited. Therefore, the fluorescence results confirm that the FLTX1 moiety efficiently transfers its energy to the RB part of the complex, which may then relax radiatively, giving place to the observed emission spectrum of FLTX2 through a Förster Resonance Energy transfer (FRET) mechanism.

### 2.6. Absorption and Fluorescence Analysis

A detailed comparison of the absorption spectra of RB and of FLTX2 reveals a change in the ratio of the main RB absorption band (longer wavelength) to its shoulder (shorter wavelength). Interestingly, the relative intensity of the main absorption band of RB to its shoulder has been considered as a measure of the aggregation of RB [43]. Shorter ratio values indicate more aggregation. In the case of RB, the ratio value is 3.85, while it decreases to 3.19 for the FLTX2 complex. This result could point to the formation of more dimmers in the FLTX2 solutions than in pure RB preparations.

In order to better understand the FRET mechanism, which underlies the fluorescent properties of FLTX2, the decays of the luminescence have been recorded. The temporal evolution of the RB emission band decay is given in Figure 5C. The decay can be fitted to an exponential decay curve:(1)$$I\left(t\right)=A\;e^{-t/\tau }$$
where τ represents the lifetime of the emission band, and A is the pre-exponential factor. The fitting was performed taking into account the reconvolution to the IRF. The best fitting was found for a lifetime constant of 2.6 ns. The decay of the fluorescence of FLTX2 when the detection is tuned at 608 nm, which corresponds to the maximum of the emission band of the RB moiety, shows a two exponential decay behavior and can be fitted to a curve:(2)$$I\left(t\right)=A_{1}e^{-t/\tau_{1}}+A_{2}e^{-t/\tau_{2}}$$
where, τ_1_ and τ_2_ are the decay constants of the fast and slow components, respectively. A_1_ and A_2_ represent the pre-exponential factors, which are related to the weight of the fast and slow components, respectively. An average life-time can be defined from the fitting parameters as [44]:(3)$$\tau_{av}=\frac{A_{1}\tau_{1}^{2}+A_{2}\tau_{2}^{2}}{A_{1}\tau_{1}+A_{2}\tau_{2}}$$

The decay constant parameters obtained from the fitting are 0.31 and 2.14 ns, for the fast and slow components, respectively. The main contribution to the decay curve corresponds to the fast component, and the average life-time is 0.51 ns, which is notably shorter than that found for the pristine RB molecules in the same solvent. There are several reasons to explain the shorter average life-time found for the RB moiety of FLTX2 as compared to the pristine RB molecules. First, non-radiative vibrational relaxation of the FLTX2 molecule might be higher than in RB molecules, as the chemical environment of the RB moiety in FLTX2 has changed. Second, the presence of dimers in FLTX2 could be higher than in pristine RB solutions, as already indicated by the absorption measurements. Since shorter decays are expected in dimers as compared to monomers, these aggregates could be partially responsible of the shortening of the average life-time. Finally, back-transfer from the RB moiety to the FLTX1 compound might take place in the FLTX2 complex. This would represent a further relaxation mechanism of the RB unit and would contribute to the shortening of its life-time. The decay of the fluorescence of FLTX1 was also measured at 550 nm upon 475 nm excitation. The decay could be fitted to a double-exponential decay, and the following fitting parameters were obtained, τ_1_ = 0.15 ns and τ_2_ = 0.54 ns. The main contribution to the decay of the fluorescence corresponds to the fast decay constant contribution. The average life-time obtained was 0.2 ns. Although the efficiency of the FRET mechanism between the FLTX1 and the RB moieties in the FLTX2 complex is high and the emission of FLTX1 moiety is almost completely quenched, a small fluorescence signal can still be detected at 550 nm, which corresponds to the FLTX1 moiety of FLTX2 (Figure 5D). Moreover, the decay of the fluorescence of this signal could be measured when FLTX2 was excited at 475 nm and the detection was tuned at 550 nm. One would expect a shortening of the life-time, as compared to pristine FLTX1, because of the new and efficient FRET relaxation mechanism to RB moiety. Surprisingly, the decay of the fluorescence of FLTX2 at 550 nm is slower than for FLTX1 [22]. It could be fitted to a double-exponential decay with τ_1_ = 0.35 ns and τ_2_ = 4.35 ns, which provide an average life-time of 2.35 ns. The fact that a longer average life-time was found for the FLTX1 moiety of FLTX2 as compared to isolated FLTX1 molecules can be understood in terms of back-transfer from the RB moiety. Indeed, the population of the excited state of the FLTX1 moiety is regulated by the longer life-time of the RB moiety, which may partially back-transfer its excited energy to populate the excited state of the FLTX1 moiety and, consequently regulates the decay rate of fluorescence.

Finally, the intramolecular FRET efficiency, E_FRET_, between the NBD and RB moieties of the FLTX2 complex can be estimated using the equation:(4)$$E_{FRET}=1-\frac{I_{550nm-FLTX2}}{I_{550nm-FLTX1}}$$
where I_550nm-FLTX2_ and I_550nm-FLTX1_ represent the emission intensities of the green emission band centered at about 550 nm (due to the NBD moieties), in the FLTX2 and FLTX1 molecules, respectively. From the experimental emission spectra of FLTX1 and FLTX2 solutions under excitation at 475 nm, the estimated efficiency (*E_FRET_*) was around 0.99, which indicates an extremely highly efficient energy transfer within the FLTX2 NBD-RB moiety. This observation strongly agrees with the negligible green fluorescence detected in the confocal images shown further on in Figure 7.

In summary, the optical spectroscopic properties found in FLTX2, as compared to those of their constituent moieties, i.e., FLTX1 and RB, allow us to conclude that an efficient energy transfer process occurs in FLTX2 complexes. Indeed, under blue excitation at 475 nm, the FLTX1 moiety of FLTX2, promotes to the excited state and transfers its energy to the RB moiety. Therefore, these optical results suggest the possible use of FLTX2 to produce ROS under blue excitation, as it is shown in the next section.

### 2.7. Laser-Stimulated ROS Generation In Vitro

In order to evaluate the potential capacity of FLTX2 as a photosensitizer, we first addressed the generation of ROS (mainly superoxide anions) using the Nitroblue tetrazolium (NBT)/formazan method depicted in Figure 6A [45].

Under this approach, the amount of oxidized formazan will depend on the photosensitizer activity of FLTX2. In our assays, we initially checked for the effect of irradiation at 473 nm, during 5 min, on the spectral response of NBT plus FLTX2 solutions in comparison with NBT plus Rose Bengal (RB) mixtures at the same concentration. In the presence of ROS, NBT transforms into formazan, which shows a characteristic broad absorption band in the 500–600 nm spectral range. The absorption spectra were recorded before and after the irradiation process. Figure 6B shows the absorption spectra of the mixtures after irradiation, after subtracting the spectra before irradiation. An absorption band in the 500–600 nm range is observed for the FLTX2 solution upon irradiation, which is attributed to formazan formation, while a negligible signal is detected in the RB alone solution. Further, the response was found to be concentration-dependent (Figure 6C). We chose the smallest concentration of FLTX2 (50 µM) to assess the effect of irradiation time on the spectral features of FLTX2. Thus, when FLTX2 samples were irradiated at 473 nm for 0, 2, 4, 6, 8, and 10 min, and the absorbance recorded between 400 and 800 nm changes in formazan generation were detected at all times above 0 min, indicating ROS generation in an irradiation time-dependent manner (Figure 6D). The increase in absorbance was totally attributable to FLTX2 molecule, since NBT, RB, or TX used alone failed to produce significant changes of absorbance in the 500–600 nm range (not shown). Further, plotting of maximal absorbance at 574 nm versus irradiation time (not shown) reveals a time-dependent increase which exhibits saturation after 8 min irradiation time.

Overall, these results indicate that, at least in vitro, irradiation of FLTX2 at the excitation wavelength of the FLTX1 moiety causes an intramolecular FRET to RB, which, in turn, undergoes formation of oxygen singlets to produce superoxide anions. These outcomes demonstrate that FLTX2 is an efficient source of ROS given proper laser stimulation. Generation of ROS may be finely tuned by adjusting irradiation times and occurs with a high FRET efficiency even in aqueous solutions. This finding is relevant from a biological point of view, and also for potential photodynamic applications.

### 2.8. Cellular Labeling of FLTX2 and Occurrence of FRET on MCF-7 Cells

MCF-7 cells were incubated with different concentrations of FLTX2 and visualized by confocal microscopy. Preparations were excited at 450 nm and the fluorescence recorded at 600 nm. Results in Figure 7A show a concentration-dependent fluorescence at the emission band of RB (600 nm). A significant fluorescence was detected in the nucleus, but most of the fluorescent signal originates at the cytoplasmic perinuclear space, both being increased as the FLTX2 concentration did. On average, cytoplasmic labeling was 3.8 times higher than that in the nucleus (Figure 7B). The evidence that FRET mechanism was responsible for the red fluorescence comes from the fact that the expected emission at 530 due to NBD moiety excitation was almost undetectable (Figure 7C), well below 5% of total cellular fluorescence (Figure 7D). The fact that most fluorescence comes from outside the nucleus disagree with the classical notion that ERs (both α and β) are essentially nuclear receptors. However, it is now widely accepted that their natural location under unstimulated conditions (i.e., in the absence of estradiol or agonist ligands) is extranuclear, being that they are found even in intracellular organelles such mitochondria and cell membranes [16,26,27,46,47].

### 2.9. Photodynamic Effects of FLTX2 on Cell Cultures

MCF-7 cells were used as a human cellular breast cancer model. We initially explored the effect of irradiation itself on untreated cells. The results shown in Figure 8A show that irradiation per se under 240 mW/cm^2^ pump power density does not affect cell viability even at exposures as long as 30 min.

In the next experiment, cells were submitted to treatment with either FLTX2 or DMSO for 2 h before irradiation using an ad hoc laser device built in our group to allow simultaneous irradiation of all wells in the plate. After 30 min irradiation, cells were incubated for 24 h under culture conditions before being processed for cell viability. Results demonstrate significant FLTX2-induced toxicity upon irradiation even at the lowest concentration assayed (1 μM), 10 μM FLTX2 being the lowest dose causing the highest mortality Figure 8B. We next assessed the effect of irradiation time on FLTX2-induced toxicity on MCF-7 cells. In these experiments we explored the effect of 10 μM FLTX2 at different irradiation times on MCF-7 cell viability (Figure 8C). We could fit the experimental data to a two-parameter exponential decay function (*p* = 0.0007, R^2^ = 0.98) which yielded a t_50_ equal 6,63 min. Thus, it is evident that irradiation time is as determinant as concentration in producing FLTX2-dependent toxicity. Finally, judging from morphological changes shown in Figure 8D, FLTX2-induced cell death likely occurs though induction of apoptosis as suggested by the signs of chromatin condensation, intracellular vacuolation (black arrows), extensive membrane blebbing (blue arrows), cell shrinkage (red arrows), altered membrane integrity, aberrant morphology, and cellular fragmentation (green arrows), which collectively represent typical damage-associated patterns of extrinsic apoptosis or type I PCD (programmed cell death) [48,49]. In agreement with our findings, previous studies in cultured cells have shown that Rose Bengal acetate photodynamic therapy (RBAc-PDT) induces exposure and release of damage-associated molecular patterns (DAMPs) [50,51]. The interpretation of these findings is that irradiated FLTX2 leads to uncontrolled generation of ROS to levels where the intracellular antioxidant systems become overwhelmed and provoke irreversible oxidative stress.

## 3. Materials and Methods

### 3.1. Synthesis of FLTX2

The synthesis of FLTX2 is schematized in Figure 1. The dimethyl acetal precursor 1 was synthesized according to a previously described method [21]. After aldehyde deprotection in acid medium, the FLTX-RB linker was introduced using a reductive amination with ethanolamine. Then, the 7-nitrobenzofurazan (NBD) moiety was attached to the amino group to yield compound 2. The esterification of alcohol 2 with Rose Bengal provided FLTX2 (3).

#### 3.1.1. Synthesis of Compound **2**

To a solution of acetal 1 (200 mg, 0.515 mmol) in THF (2.5 mL) was added a 3M HCl aqueous solution (2.5 mL). The reaction was stirred for 6 h at 50 °C. The mixture was then allowed to cool to room temperature, diluted with water (50 mL) and extracted with EtOAc (3 × 50 mL). The joint organic phases were dried over MgSO_4_ and concentrated under vacuum. The resulting crude aldehyde (not shown) was dissolved in THF (2 mL) under nitrogen atmosphere and stirred with magnesium sulphate (1 g) and ethanolamine (60 μL, 1.0 mmol) for 18 h. The reaction was then filtered and concentrated under vacuum. The crude oily product was dissolved in ethanol (5 mL), treated with NaBH_4_ (46 mg, 1.2 mmol) and stirred at room temperature for 4 h. The reaction was quenched with a 5% HCl aqueous solution, extracted with EtOAc, dried over MgSO_4_, and concentrated under vacuum.

The resulting amine was dissolved in CH_2_Cl_2_ (5 mL) and treated with triethylamine (0.25 mL) and 4-chloro-7-nitrobenzofurazan (NBD-Cl, 100 mg, 0.5 mmol). The reaction was stirred for 4 h at room temperature and then concentrated under vacuum. The residue was purified by column chromatography (hexane/EtOAc, from 9:1 to 1:1) to give compound 2 as a yellow solid (81 mg, 29% for the four steps). 

**^1^****H NMR** (500 MHz, CDCl_3_, isomer mixture) δ 8.40–8.27 (m, 1H), 7.50–7.07 (m, 8H), 7.05–6.98 (m, 2H), 6.98–6.96 (m, 2H), 6.86–6.82 (m, 1H), 6.77 (d, *J* = 8.8 Hz, 1H), 6.50 (d, *J* = 8.8 Hz, 1H), 6.35/6.27 ([d, *J* = 9.1 Hz/ d, *J* = 9.0 Hz], 1H), 4.55–4.50 (m, 1H), 4.44–4.38 (m, 1H), 4.38 (t, *J* = 4.9 Hz, 1H), 4.31–4.25 (m, 1H), 4.22 (t, *J* = 5.0 Hz, 1H), 4.21–4.16 (m, 1H), 4.13 (t, *J* = 5.2 Hz, 1H), 4.06 (t, *J* = 5.2 Hz, 1H), 2.45/2.44 ([ddd, *J* = 8.6, 7.9, 7.6 Hz/ ddd, *J* = 8.6, 7.6, 7.4 Hz], 2H), 0.92/0.91 ([t, *J* = 7.5 Hz/ t, *J* = 7.5 Hz], 3H).

**^13^****C NMR** (126 MHz, CDCl_3_, isomer mixture). The isomer signals corresponding to the same carbon are separated by a dash. Some key signals are described) δ 156.6/155.7 (=C-O), 145.4 (C), 144.7/144.6 (C), 143.6/143.1 (C), 142.31/142.26 (C), 142.1/141.9 (C), 138.0/137.9 (C), 137.1/136.6 (C), 135.2 (C), 132.1, 130.8, 130.7, 129.7, 129.6, 129.4, 128.2, 127.9, 127.8, 127.4, 126.6, 126.2, 126.1, 125.8, 122.8 (C, C-NO_2_), 114.0/113.2 (CH, CH=C-O), 102.3/102.2 (CH, NBD), 65.9 (CH_2_O), 63.4/60.3 (CH_2_O), 56.9/56.8 (CH_2_N), 54.2/54.0 (CH_2_N), 31.9/29.0 (CH_2_), 13.5/7.7 (CH_3_).

**HRMS-ESI:** Calcd for C_32_H_31_N_4_O_5_ [M + H]^+^: 551.2294, found: 551.2273. 

**Elemental Analysis** calcd for C_32_H_30_N_4_O_5_: C, 69.80%; H, 5.49%; N, 10.18%; found: C: 69.81%, H: 5.52%, N: 9.98%.

**^1^****H NMR and ^13^C NMR** spectra can be found in the Appendix A (Appendix A).

#### 3.1.2. Synthesis of FLTX2 (Compound **3**)

Compound 2 (81 mg, 0.15 mmol) was dissolved in dry DMF (5 mL) under nitrogen atmosphere, and the solution was treated with HBTU (62 mg, 0.16 mmol), DMAP (20 µL, 20 mg, 0.16 mmol), Rose Bengal (167 mg, 0.16 mmol), and DIPEA (0.78 mL, 4.5 mmol). The reaction mixture was stirred at room temperature for 18 h, and then was concentrated under vacuum. The residue was purified by column chromatography (EtOAc) to give FLTX2 (3) (93 mg, 42%) as a red solid.

**^1^****H NMR** (500 MHz, acetone-d_6_, isomer mixture) δ 8.54/8.50 ([d, *J* = 9.0 Hz/ d, *J* = 9.0 Hz], 1H), 7.69/7.66 (s/s, 1H), 7.38 (t, *J* = 7.7 Hz), 7.30–7.08 (m, 8H), 7.04–6.95 (m, 1H), 6.89 (d, *J* = 9.0 Hz, 3H), 6.77 (d, *J* = 8.8 Hz), 6.54 (d, *J* = 8.8 Hz), 6.49/6.41 ([d, *J* = 9.0 Hz/ d, *J* = 9.0 Hz], 1H), 4.48–4.44 (m, 1H), 4.44 (t, *J* = 6.0 Hz), 4.42–4.36 (m, 2H), 4.37–4.32 (m, 1H), 4.27–4.20 (m, 2H), 4.19–4.13 (m, 1H), 2.49–2.37 (m, 2H), 0.90–0.77 (m, 3H).

**^13^****C NMR** (126 MHz, acetone-d_6_, mixture of isomers. Some isomer signals corresponding to the same carbon are separated by a dash). δ 171.9 (C, CO/COCH=C(O)), 163.1 (C, CO_2_), 157.72 (C), 157.70 (C), 157.2 (C), 156.4 (C), 145.10/144.97 (C), 143.7, 143.4, 142.3, 141.3, 139.4, 138.6, 138.4, 136.7, 136.6, 135.7, 135.5, 134.7, 134.1, 132.6, 131.7, 130.6, 130.4, 129.7, 129.6, 129.2, 128.1, 127.9, 127.3, 126.5, 126.0, 125.6, 114.2 (CH), 113.4 (CH), 110.80/110.77 (=C), 103.11/103.03 (CH, NBD), 96.4 (C, =C-I), 75.6 (2×C, =C-I), 65.7/65.5 (CH_2_O), 63.3 (CH_2_O), 54.1 (C, =C-I), 53.7/53.6 (CH_2_N), 51.9/51.8 (CH_2_N), 12.92/12.86 (CH_3_). An aliphatic CH_2_ signal is overlapped with the solvent.

**HRMS-ESI**: Calcd for C_52_H_31_Cl_4_I_4_N_4_O_9_ [M – H]^−^: 1502.7024; found: 1502.6992. 

**Elemental Analysis:** calcd for C_52_H_32_Cl_4_I_4_N_4_O_9_: C, 41.46%; H, 2.14%; N, 3.72%; found: C, 41.65%; H, 1.93%; N, 3.45%.

**^1^****H NMR and ^13^C NMR** spectra can be found in the Appendix A (Appendix A), as well as DEPT (Appendix A) and HSQC (Appendix A) experiments.

### 3.2. Estrogen Receptor Competitive Binding Assay

Enriched preparations of Estrogen receptor were obtained from uterine cytosol fraction from mature female Sprague-Dawley rats following the procedure described elsewhere [16]. Aliquots of 100 µL cytosol were incubated with 5 nM [^3^H]E2 and increasing concentrations of non-radioactive FLTX2 or TX (0.1 nM–100 µM) for 18 h at 4 °C. Then, 200 µL of dextran (0.08%)-coated charcoal (0.8%) suspension prepared in TRIS-EDTA-Glycerol-Mg buffer was added to each tube and incubated for 10 min. Suspensions were then centrifuged for 10 min at 3000 g. Radioactivity was then measured in the supernatant in 4 mL scintillation cocktail Optiphase Hisafe 2 (PerkinElmer) by LKB Rackbeta counter (LKB Instrument). Corrections were made for non-specific binding. Relative binding affinity (RBA) was calculated as the ratio of FLTX2 and TX IC_50_ values obtained from dose-response curves.

### 3.3. Docking Studies

In order to get a deeper insight about the higher affinity of FLTX2 over tamoxifen on ERα, we initially performed in silico docking studies. The X-ray coordinates of human ERα ligand binding domains (LBD) were extracted from the Protein Data Bank (PDB code 3ERT). The PDB structures were prepared for docking using the Protein Preparation Workflow (Schrodinger, LLC, New York, NY, USA, 2020) accessible from the Maestro program (Maestro, version 12.3; Schrodinger, LLC: New York, NY, USA, 2020). The substrate and water molecules were removed beyond 5 Å, bond corrections were applied to the cocrystallized ligands and an exhaustive sampling of the orientations of groups was performed. Finally, the receptors were optimized in Maestro 12.3 by using OPLS3e force field before docking study. In the final stage the optimization and minimization on the ligand-protein complexes were carried out with the OPLS3e force field and the default value for RMSD of 0.30 Å for non-hydrogen atoms were used. The receptor grids were generated using the prepared proteins, with the docking grids centered at the bound ligand for each receptor. A receptor grid was generated using a 1.00 van der Waals (vdW) radius scaling factor and 0.25 partial charge cutoff. The binding sites were enclosed in a grid box of 20 Å^3^ without constrains. The three-dimensional structures of the ligands to be docked were generated and prepared using LigPrep as implemented in Maestro 12.3 (LigPrep, Schrodinger, LLC: New York, NY, USA, 2020) to generate the most probable ionization states at pH 7 ± 1 (retaining the original ionization state). These conformations were used as the initial input structures for the docking. In this stage a series of treatments were applied to the structures. Finally, the geometries were optimized using OPLS3e force field. These conformations were used as the initial input structures for the docking. The ligands were docked using the extra precision mode (XP) [52] without using any constraints and a 0.80 van der Waals (vdW) radius scaling factor and 0.15 partial charge cutoff. The dockings were carried out with flexibility of the residues of the pocket near to the ligand. The generated ligand poses were evaluated with empirical scoring function implemented in Glide, GlideScore, which was used to estimate binding affinity and rank ligands [53]. The XP Pose Rank was used to select the best-docked pose for each ligand.

### 3.4. Molecular Dynamics Simulation

Optimized Potentials for Liquid Simulations-2005 (OPLS2005) [54] force field in Desmond Molecular Dynamic System was used in order to study the behavior of the ligand-target complex. The docking resulting complexes were solvated with an orthorhombic box of TIP3P (Transferable Intermolecular Potential 3-Point) water [55] and counter ions were added, creating an overall neutral system simulating approximately 0.15 M NaCl. The ions were equally distributed in a water box. The final system was subjected to a MD simulation up to 50 ns using Desmond program [56]. The method selected was NPT (Noose-Hover chain thermostat at 300 K, Martyna-Tobias-Klein barostat method at 1.01325 bar with a relaxation time of 2 ps, isotropic coupling, and a 9 Å radius cut-off was used for coulombic short-range interaction) constraints were not applied. During the simulations process, smooth particle Mesh-Ewald method was used to calculate long-range electrostatic interactions. For multiple time step integration, RESPA (Reversible reference System Propagator Algorithm) was applied to integrate the equation of motion with Fourier-space electrostatics computed every 6 fs, and all remaining interactions computed every 2 fs [57]. MD simulations were carried out on these equilibrated systems for a time period of 50 ns, frames of energy and trajectory were captured after every 1.2 ps and 9.6 ps, respectively. The quality of MD simulations was assessed by the Simulation Event Analysis tool. Ligand-receptor interactions were identified using the Simulation Interaction Diagram tool.

### 3.5. Optical Measurements

A continuous wave (CW) 473 nm diode laser was used in the laser irradiation studies. The collimated beam of the laser was expanded using two convergence lenses to obtain a 1 cm diameter collimated beam. The irradiation density was 240 mW/cm^2^. In the in vitro experiments, the frontal face of a quartz cuvette was irradiated, providing a homogenous excitation of the solutions. In the case of the in vivo studies, a mirror was utilized to obtain a homogeneous illumination from the top of the cell culture, which completely covered the cell culture well.

Optical absorption in the visible spectral range was performed in an Agilent Cary 5000 spectrophotometer equipped with double beam cuvette holders. For fluorescence studies, an Edinburgh Instruments LifeSpec II fluorescence spectrometer was used. An Edinburgh Instruments EPL-475 picosecond pulsed diode laser (typical temporal pulse width 80 ps) was selected as the excitation source to pump samples at 475 nm. Fitting of decay curves to mono- or double-exponential equations was made using instrumental response function (IRF) reconvolution analysis with Edinburgh Instruments FAST software. All measurements were conducted at room temperature.

### 3.6. ROS Generation

The potential capacity of FLTX2 as a photosensitizer, and its ability to produce reactive oxygen species, was determined using a colorimetric assay based on the nitroblue tetrazolium (NBT)/formazan method [45]. NBT is a yellow, water-soluble compound that is transformed into dark-blue formazan upon reaction with superoxide ions and other reactive oxygen species. NBT-formazan displays a broad absorption band at 500–600 nm whose intensity can be measured [20,58]. The amount of formazan is thus directly related to photosensitizer activity of FLTX2.

NBT tablets were purchased from Sigma Aldrich and dissolved in deionized water. One mL solutions of 50 μM FLTX2 (in DMSO) and 100 μM NBT (final concentrations) were prepared and irradiated with a CW flux of laser radiation at 473 nm at a power density of about 240 mW/cm^2^ during 0, 4, 8, 12, 16, and 20 min. Control solutions without FLTX2 were also prepared and irradiated at the same times and control mixtures. Samples were assessed spectrophotometrically in the range 400–800 nm and the absorbance, after subtraction at t = 0, was plotted as a function of wavelength and irradiation times.

### 3.7. Cellular Culture

MCF-7 cells were grown in standard DMEM culture medium containing 10% FBS and maintained at 37 °C under 95% air/5% CO_2_ atmosphere. Upon reaching 90% confluency, cells were detached from the flasks with trypsin-EDTA and seeded depending on the subsequent use, i.e., viability/toxicity assays or fluorescence analyses.

### 3.8. FLTX2-Induced Cellular Toxicity

For viability assays, MCF-7 cells were seeded at a density of 25.000 cells per well in chambered cell culture slides (Falcon). Cells were grown for 72 h and after this time, media was replaced with a fresh one containing the corresponding dose of FLTX2 and incubated for 2 h. After incubation, cells were irradiated for 5, 10, 15, or 30 min with the 473 nm diode laser and irradiation power density was 240 mW/cm^2^ as previously described. Corresponding non-irradiated control was performed using the opposite well to the irradiated one in the same slide. After 24 h irradiation, 30 µL of Trypan Blue 0.4% was added and pictures from 5 random fields per well were taken. Experiments were performed by triplicated. Dead cells are easily recognizable by their trypan blue staining and their tendency to be floating. Viability was automatically measured using the cell counter application from ImageJ software.

### 3.9. Confocal Microscopy Studies

Cellular fluorescence studies were performed following the procedure described in Morales et al., (2016) [46]. Briefly, MCF7 cells were seeded in 8-well chamber slides (40,000 cells/well) and maintained for 24 h at 37 °C in a 95% air/5% CO_2_ atmosphere. For fixation, cells were incubated for 1 min with the fixative solution (2% paraformaldehyde, 0.1% glutaraldehyde, 150 mM sucrose), followed by 2 min in 0.5% Nonidet P-40 solution at room temperature. After fixation, cells were washed three times with PBS and incubated in BSA 5% (30 min) to block background noise.

Once washed, cells were incubated for 2 h with FLTX2 (50 or 100 µM), dissolved in DMSO (<2%), and washed again with PBS. A drop of mounting solution containing glycerol and DAPI was applied to each well and the slide was covered with a coverslip and analysed by confocal microscopy (Leica SP8 with software LAS X from Leica). Preparations were excited at 450 nm and the fluorescence recorded at 530 nm and 600 nm. Fluorescence quantification was done using ImageJ software (Rasband, W.S., ImageJ 1997–2018. https://imagej.nih.gov/ij/ accessed on 2–15 July 2020). Fluorescence from the nucleus and cytoplasm were measured independently. Corrected Total Cell Fluorescence (CTCF) was calculated for each compartment as:(5)CTCF=Id−(CA∗Bkg)
where CTCF is the corrected total cell fluorescence, Id is the integrated fluorescence density, CA is the cell area, and Bkg is the mean background fluorescence [46].

### 3.10. Statistical Analysis

When required, data were submitted to one-way ANOVA test followed by Tukey’s post hoc test, or by non-parametric Kruskal Wallis test followed by Games-Howell’s post hoc. Student-Newman-Keuls *t*-test was also used to determine differences between treatments or times. Dose-response curves were fitted to four parameters logistic equation using nonlinear regression analysis tools included in the statistical software.
(6)$$\%Binding=min+\frac{max-min}{1+{\left(\frac{x}{EC50}\right)}^{nH}}$$
where EC50 is the concentration producing 50% of total binding, nH is the Hill coefficient, and x is the experimental binding.

## 4. Conclusions

The newly developed FLTX2 is a compound that is pharmacologically and optochemically active. FLTX2 molecule maintains the triphenylethylene core of TX essential to bind the LBD pocket of ER but the lateral side formed by the NBD-linker-RB moiety protrudes out the LBD and displaces helix 12 from adopting its agonist conformation. Experimentally, FLTX2 retains the antiestrogen potency of TX in breast cancer cell lines, and is likely to be devoid of estrogenic effects as it was demonstrated for its predecessor FLTX1. Fluorescence studies demonstrate an efficient intramolecular FRET mechanism in FLTX2. Moreover, under blue light excitation, the FLTX1 moiety of the complex efficiently transfers its excited energy to the RB moiety through a FRET mechanism. RB excited states can then relax, transferring its energy to oxygen ground state triplets to induce ROS. Indeed, FLTX2 exhibits a highly efficient ability to generate superoxide radicals and to induce cell death, likely through ROS-induced apoptosis. Consistent with the high intramolecular FRET efficiency of FLTX2, nearly all fluorescence in fixed MCF7 cells is observed in the emission band of RB. Further, most fluorescent signals originate at the cytoplasmic perinuclear space, as expected for unstimulated MCF-7 cells.

Overall, these properties of FLTX2 make this derivative a potential heir of TX, as a pharmacophore endowed with the ability to specifically bind ER and to antagonize ER activation in estrogen-dependent breast cancer cells, which upon appropriate irradiation would undergo uncontrolled oxidative stress.

## Figures and Tables

**Figure 1 ijms-22-05339-f001:**
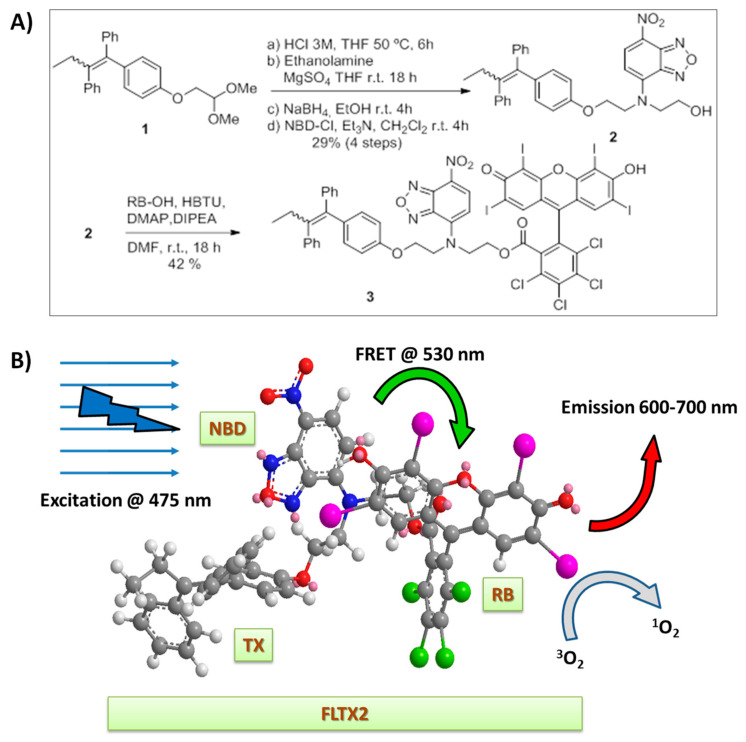
(**A**) Synthesis of FLTX2 (compound 3). (**B**) Minimized 3D representation of FLTX2 (3) highlighting the sequence: excitation wavelength, intramolecular FRET, red emission, and singlet oxygen generation. For details see optical measurements DIPEA: Diisopropylethylamine; DMAP: Dimethylaminopyridine; DMF: Dimethylformamide; HBTU: O-(Benzotriazol-1-yl)- *N,N,N′,N′*-tetramethyluronium hexafluorophosphate; NBD: 7-nitrobenzofurazan; THF: tetrahydrofuran.

**Figure 2 ijms-22-05339-f002:**
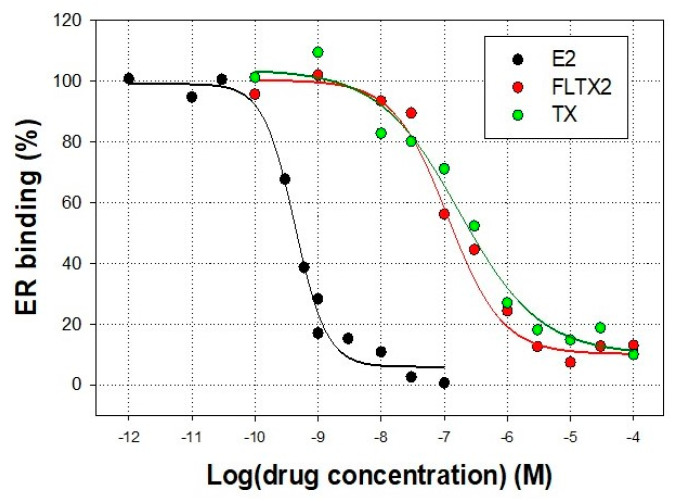
Effects of FLTX2 and TX on [^3^H]-estradiol competitive ER binding assay. Uterine cytosolic extracts were saturated with 5 nM of labelled estradiol in the presence of increasing concentrations of unlabelled estradiol (E2), tamoxifen (TX), or FLTX2. Data are presented as mean ± SEM of four different assays.

**Figure 3 ijms-22-05339-f003:**
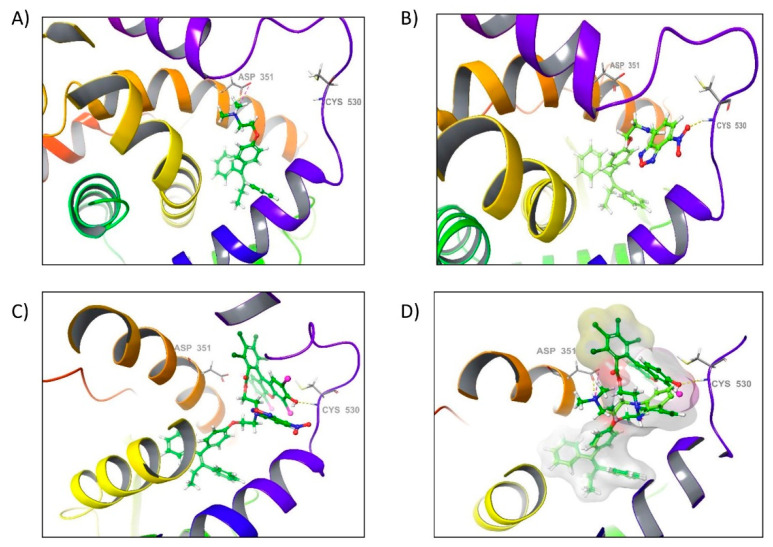
Best Docking poses for TX (**A**), FLTX1 (**B**), and FLTX2 (**C**) in the human ERα ligand binding domains (LBD). (**D**) Overlapped docking poses of TX and FLTX2 within the human ERα LBD.

**Figure 4 ijms-22-05339-f004:**
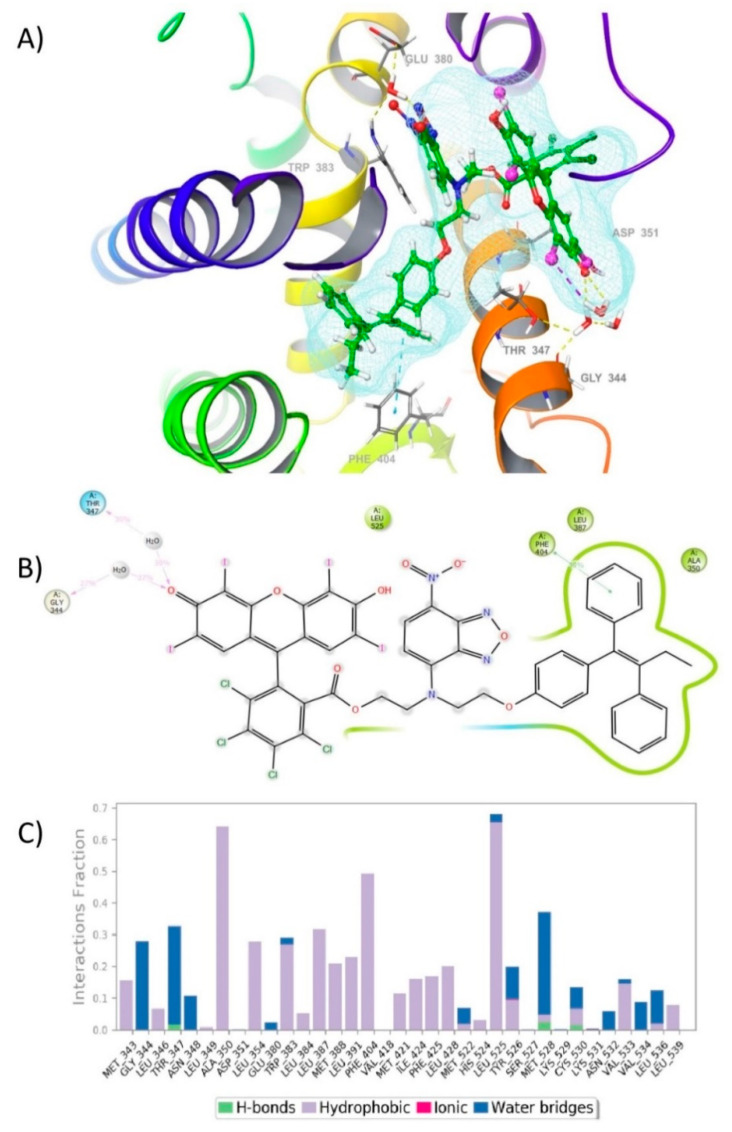
Molecular Dynamics study for FLTX2 on the human ERα ligand binding domain. (**A**) Interactions of FLTX2 with key amino acid residues at the hydrophobic binding pocket of human ERα LBD. (**B**) 2D Ligand-protein interaction diagrams and interaction strength quantified by the frequency of occurrences in the trajectory when a minimum percentage of 50% is achieved. Colors indicate type of residue: green are lipophilic residues; blue are polar residues; purple are basic residues. Ligand atoms that are exposed to the solvent are marked with grey spheres. (**C**) Stacked bar charts of human ERα LBD interaction with FLTX2.

**Figure 5 ijms-22-05339-f005:**
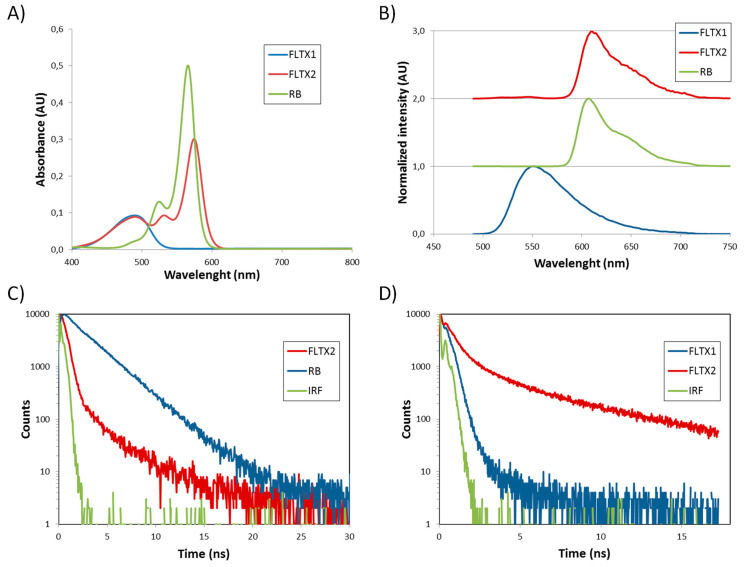
Optical properties of FLTX2. (**A**) Absorption spectra of FLTX1, RB, and FLTX2. (**B**) Normalized emission spectra of FLTX1, RB, and FLTX2. (**C**) Decay of the fluorescence of RB and of FLTX2 with detection tuned at 608 nm. (**D**) Decay of the fluorescence of FLTX1 and of FLTX2 with detection tuned at 550 nm.

**Figure 6 ijms-22-05339-f006:**
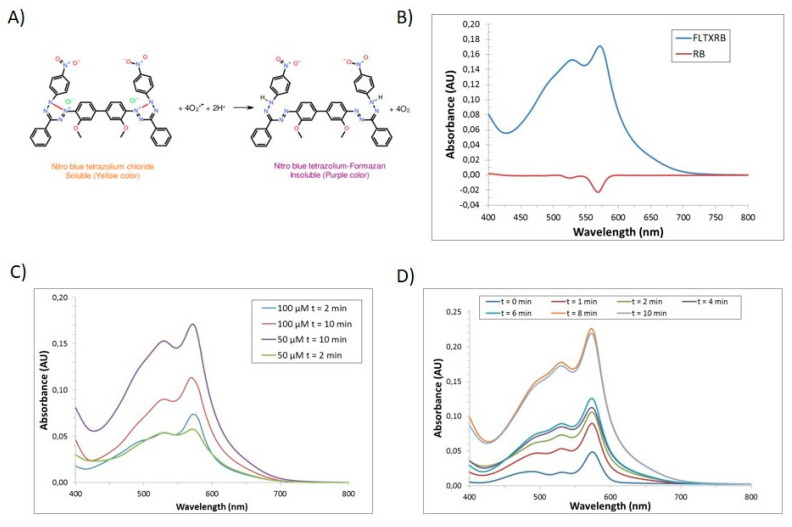
FLTX2-induced ROS generation. (**A**) Superoxide anion-induced formation of nitro blue tetrazolium-formazan. (**B**) Absorption spectra of equimolar FLTX2 and RB solutions after laser irradiation for 10 min. (**C**) Concentration-dependence of FLTX2-induced formazan formation irradiated for 2 or 10 min. (**D**) Time-dependence for FLTX2-induced (50 µM) formazan generation.

**Figure 7 ijms-22-05339-f007:**
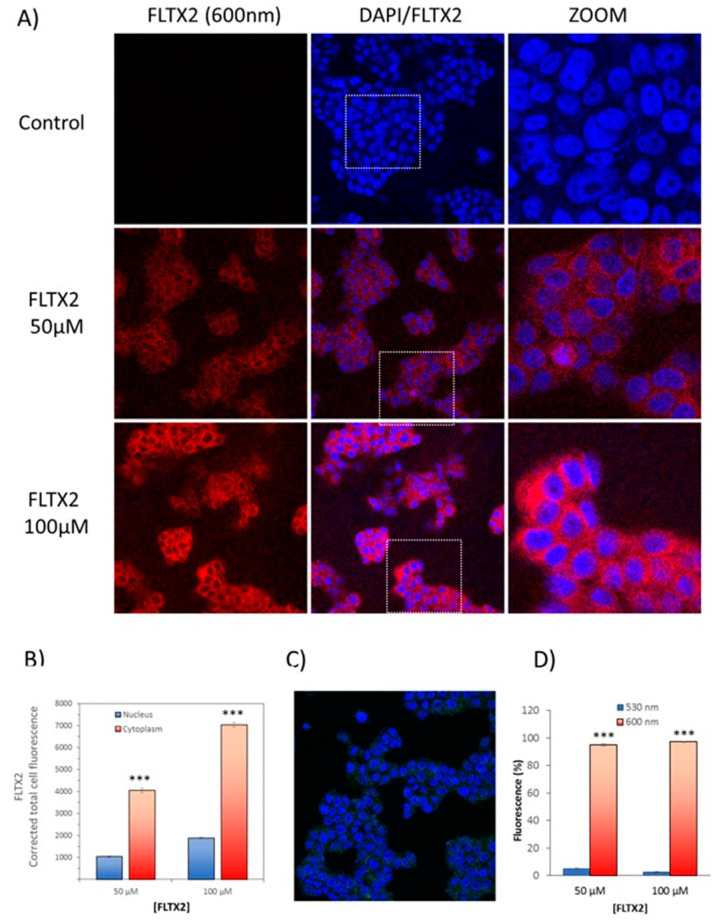
Cellular labeling of FLTX2 and occurrence of FRET in MCF-7 cells. (**A**) Representative images of irradiated preparations (450 nm, 30 min) incubated with different concentrations of FLTX2 (0, 50, and 100 μM). Fluorescent signals were recorded at the emission band of RB (600 nm). DAPI was used as nuclear marker. (**B**) Concentration-dependent fluorescent signals in cytoplasmic and nuclear compartments. (**C**) Demonstration of FRET efficiency in FLTX2 irradiation. Fluorescent signal was virtually absent at the emission wavelength of the NBD moiety of FLTX1. (**D**) bar chart comparing fluorescence signals recorded at 530 nm (NBD moiety) and 600 nm (NBD to RB transfer). *** Statistically significant at *p* < 0.005.

**Figure 8 ijms-22-05339-f008:**
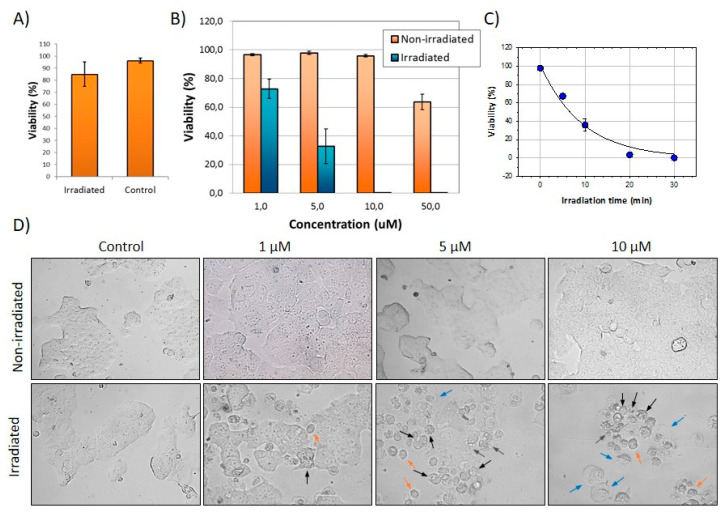
Photodynamic effects of FLTX2. (**A**) Effects of laser irradiation (475 nm) for 30 min on MCF-7 cells viability. (**B**) Concentration-dependence effects on MCF-7 cells viability irradiated for 30 min. (**C**) Effects of irradiation time on FLTX2-induced toxicity. (**D**) Phase contrast transmission representative images of irradiated and non-irradiated MCF-7 cells at different concentrations of FLTX2. Arrows indicate different apoptotic traits: intracellular vacuolation (black arrows), membrane blebbing (blue arrows), cell shrinkage (red arrows), cellular fragmentation (green arrows).

## Data Availability

The datasets generated during and/or analysed during the current study are available from the corresponding author on reasonable request.

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
