# Peer review of "FLTX2: A Novel Tamoxifen Derivative Endowed with Antiestrogenic, Fluorescent, and Photosensitizer Properties"

_ijms, 2021, doi:10.3390/ijms22105339_

Round 1

Reviewer 1 Report

  1. Discussion and introduction, please implement current information 2021-2011
  2. Conclusion,please rewrote

Author Response

Dear reviewer,

Thank you for your brief comments. We understand you reckon we should have used more recent references in Results and Discussion section. Accordingly, we have updated our reference list and modified the text accordingly. 

Conclusion section was changed.

Reviewer 2 Report

In my opinion this paper entitled "FLTX2: A Novel Tamoxifen Derivative Endowed with Antiestrogenic, Fluorescent and Photosensitizer Properties" submitted to International Journal of Molecular Sciences seems to be rather useful and quite interesting. I find the research designed properly. The advantages of FLTX2 over FLTX1 - being of great importance - were described and are obvious. The paper is generally properly written, and the quality of the text is rather good (some grammatical and typographical errors occur). In my opinion the manuscript is worthy enough to be published in International Journal of Molecular Sciences after incorporating of some minor changes (see below).

  1. In my opinion the text in section 2.1 (results and discussion) should be in a large part transferred to section 3.1 (materials and methods).
  2. For equations 5 and 6 appropriate references should be added.
  3. In line 27 the abbreviation NBD should be explained.
  4. In line 28 there should be “demonstrate” instead of “demonstrates”
  5. In line 42 there should be “triphenylethylene” instead of “triphenylethilene”.
  6. In line 64 there should be “photosensitizer” instead of “photosentitizer”.
  7. Once there is “opto-chemical”, “reactive oxygen species”, “7-nitrobenzofurazan”, another time “optochemical”, “reactive-oxygen species”, “nitrobenzofurazan”. Please correct and unify.
  8. Lines 77-79 should be removed.
  9. In line 104 there should be “wavelength” instead of “wavelength”.
  10. Please check in the whole paper sub- and superscripts (see for example line 614).

Author Response

Dear reviewer,

Thank you very much for your pertinent comments and detailed examination of our manuscript. Following your suggestions, we have corrected and modified all points you have mentioned, all of them highlighted in yellow in the new version. 

With regards to your first comment, we have placed most information and technical aspects regarding the synthesis of FLTX2 in the Materials and Methods section. The purpose to introduce FLTX2 in this section is that it is a newly developed molecule tailored under a paradigm of incorporating new functionalities to a well-known anticancer drug: tamoxifen. This is unprecedented and a central issue in the current study. Therefore, we have introduced a brief paragraph with the rationale for the synthesis of FLTX2 as well as short methodological issues such as intermediates and yields, which are of particular interest for researchers in organic chemistry. Synthesis details are, as you find appropriate, in Materials and Methods section.